# The Profile and Development of the Lower Limb in Setswana-Speaking Children between the Ages of 2 and 9 Years

**DOI:** 10.3390/ijerph17093245

**Published:** 2020-05-06

**Authors:** Mariaan van Aswegen, Stanisław H. Czyż, Sarah J. Moss

**Affiliations:** 1Physical Activity, Sport and Recreation (PhASRec), Faculty of Health Sciences, North-West University, Potchefstroom 2525, South Africa; 20383800@nwu.ac.za (M.v.A.); Hanlie.moss@nwu.ac.za (S.J.M.); 2Department of Sport Didactics, University School of Physical Education in Wrocław, 51-612 Wrocław, Poland; 3Incubator of Kinanthropology Research, Faculty of Sport Studies, Masaryk University, 625 00 Brno, Czech Republic

**Keywords:** lower limb development, tibiofemoral angle, quadriceps-angle, hip anteversion angle

## Abstract

Profile data on normal lower limb development and specifically tibiofemoral angle development in black, Setswana-speaking South African children are lacking. This study aimed to provide profiles on the development of the tibiofemoral angle, hip anteversion angle and tibial torsion angles in two- to nine-year-old children. Measurements of the tibiofemoral angle, intercondylar distances or intermalleolar distances, quadriceps-angle, hip anteversion- and tibial torsion angle were clinically obtained from 691 healthy two- to nine-year-old children. Two-year-old children presented with closest to genu varum at −3.4° (±3.4°). At three years, a peak of −5.7° (±2.3°) genu valgum was seen, which plateaued at −4.5° (±2.1°) at age nine years. Intermalleolar distance results support tibiofemoral angle observations. Small quadricep-angles were observed in the two-year-old group, (−3.81° ± 3.77°), which increased to a mean peak of −9.2° (±4.4°) in nine-year-olds. From the age of four years old, children presented with neutral tibial torsion angles, whilst two- and three-year-olds presented with internal tibial torsion angles. Anteversion angles were the greatest in three-year-olds at 77.6° ± 13.8° and decreased to a mean angle of 70.8° ± 6.9° in nine-year-olds. The tibiofemoral angle developed similarly to those tested in European, Asian and Nigerian children, but anteversion- and internal tibial torsion angles were greater in the Setswana population than angles reported in European children. Our findings indicate that lower limb development differs in different environments and traditions of back-carrying may influence the development, which requires further investigation.

## 1. Introduction

Knee development profile information is vital to healthcare providers, e.g., pediatricians and physical therapists, while assessing child knee alignment [1,2,3]. The development of the tibiofemoral angle (TFA), previously described in American [4], Chinese [1], Indian [3], and Nigerian cohorts [5,6], is used to investigate knee alignment [1,3,4,5,6,7,8,9,10,11,12,13,14]. In normal children, there is a progression of the knee angle, which varies mildly between children from different ethnicities [1,4,9,10,12,15,16,17]. Infants present with knee varus angles (bow legs) shortly after birth, which changes to a valgus angle (knock-knees) in early childhood [1,3,4,8,9,12]. In addition to TFA, intercondylar distance (ICD) for genu varum (distance between the medial femoral condyles) or the intermalleolar distance (IMD) for genu valgum (distance between medial tibial malleoli) [1,3,4,5,6,7,8,9,10,17], is often used to assess knee alignment. Tibia vara, internal tibial torsion (TT) and femoral retroversion often accompany genu varum [18]. This necessitates the inclusion of whole kinetic chain measurements, from the hip to the ankle, including both frontal- and transverse-plane observations, such as femoral anteversion (AV) and TT. The quadriceps-angle (Q-angle) indicates the muscular lateral pull of the quadriceps’ musculature and may be indicative of femoral AV [19]. Differences in Q-angle measurement protocol can lead to significant variation in measurements [19,20]. Despite the limitations, the field Q-angle measure is still valuable. 

In South Africa, a low-middle income country, where over- and undernutrition exist within the same population [21], the back-carrying of children is a culturally accepted norm. Against the background of potentially poor nutrition at the onset of life, profiling lower limb (LL) development and the influence of back-carrying on LL development, would be of interest because current data are lacking. Setswana speakers (approximately 12 million) reside in the larger central region of South Africa and Botswana, but the language is also spoken in parts of Namibia, Zimbabwe, and Lesotho [22]. This study forms part of a larger study where LL development of back-carried Setswana-speaking children will be compared to the LL development of non-back-carried children. This comparison of LL development in Setswana-speaking children requires normative data of LL development. It will not be sufficient to compare LL development of Setswana-speaking children to the development reported in children of other ethnicities, due to researchers finding differences in LL development in children of different ethnicities [1,3,4,5,6,7,8,9,10,11,12,13,14]. In addition, healthcare providers in South Africa can utilize our data on LL development for Setswana-speaking children rather than those for children of other ethnicities when evaluating their patients. The aim of this study is to profile the normal development of the LL in terms of the TFA, IMD or ICD, Q-angle, anteversion angle (AVA), and tibial torsion angle (TTA), in Setswana-speaking, South African children between 2 and 9 years of age. Age-, limb- and sex-related analyses were performed for all parameters, only significant results were included.

## 2. Materials and Methods 

The study was approved by the Ethics Committee (NWU – 00094 17 A1). All participants and their legal guardians voluntarily signed (if able to) assent and informed consent, respectively.

### 2.1. Participants 

The 1382 LL measurements were taken from 691 (Table 1), Setswana-speaking children between the ages of two- and nine years. We aimed to recruit 100 children per age category, to allow percentile ranking; however, recruitment of the younger age categories, between ages two and four years, proved challenging. Recruitment was conducted at crèches and primary schools in Potchefstroom and Ikageng, North-West Province, South Africa, listed and registered by the Department of Education and the Department of Social Development. Of the potential 18 crèches and 24 primary schools in Potchefstroom and Ikageng, six schools and eight crèches agreed to participate. Typically, primary schools have approximately 180 learners per grade. This renders a total population of 3240 potential participants, across three grades (excluding grade R, since many primary schools did not have a grade R). Principals of ten schools declined participation, based on the lack of Setswana-speaking children attending their schools. The crèches visited typically are much smaller, with a total of approximately 50 to 80 children attending a crèche. Children were grouped to the nearest integer, according to their chronological age. All participants were Setswana-speaking, healthy, with no known history of musculoskeletal disorders and could walk. A total of 724 children were tested, of which 33 were excluded for reporting musculoskeletal disorders in a self-report questionnaire.

### 2.2. Procedure 

A self-report questionnaire was utilized to determine inclusion and exclusion criteria. Before measurements, all anatomic landmarks were identified through palpation and marked. The measurements were performed by the main researcher (M.v.A.), while trained assistants ensured the subject maintained neutral hip rotation and full knee extension. TFA testing considered the angle between the line from the anterior superior iliac spine (ASIS) to the center of the patella and the line from the center of the patella to the center of the ankle joint [1,3]. Varus angles were reported in positive degrees and valgus angles as negative degrees, as measured with a goniometer. The ICD was measured when the child presented with a varus alignment, as the distance between the femoral condyles, whilst the child was standing with his/her hips and knees in neutral alignment and extended, and the ankles touching. IMD was similarly measured for a child presenting with a valgus alignment as the distance between the medial malleoli and with the knees touching [1,3,6].

The AVA or retroversion angle was determined clinically through Craig’s test [19], where the child lay in prone with the knee flexed to 90°. The position where the greater trochanter projected most laterally and parallel to the plinth surface upon internal and external hip rotation was held. The angle between the true vertical and the lower leg was measured. To allow accurate measurement, the angle between the horizontal border of the plinth and the central axis of the LL was measured and subtracted from 90°. 

The Q-angle was identified as the angle between a line from the ASIS to the center of the patella and a line from the center of the patella to the tibial tuberosity. The Q-angle was obtained with the child standing, knees in extension, quadriceps musculature relaxed and the feet and hips in neutral alignment [19]. Measurements of body mass, height, and body mass index were included for descriptive statistical purposes (Table 1) [23,24].

### 2.3. Statistical Analysis 

Descriptive statistics, including means, standard deviations (SD), medians and 25th and 75th percentiles (Q1 and Q3 respectively) were reported. The differences between left and right TFAs, IMDs, Q-angles, TTAs and AVAs of the participants were assessed, using *t*-tests with Bonferroni adjustments, i.e., original p-values were multiplied by six, since six measurements (IMD, ICD, AVA, TFA, TTA and Q-angle) were taken in each age group and six t-tests were performed on the same population (males and females). Cohen’s d was used to estimate effect size. Limb difference comparison were performed for both sexes and per age group for all parameters except for IMDs which cannot distinguish limb discrepancies. Where no significant differences were found between sexes or limbs, data were pooled and reported as overall means per age group.

## 3. Results

### 3.1. Tibiofemoral Angle (TFA) and Intermalleolar Distance (IMD)

Significant differences were observed between males and females at certain ages and between limbs in terms of the TFA. The mean TFA in 2-year-olds was −3.39° ± 3.43° and the closest to varus angulation of all the age groups (Table 2). Peak valgus was observed in 3-year-olds, with a mean TFA of −5.63° ± 2.26° with −11° as the minimum and −0.5° as the maximum. The TFA progressed from valgus towards more neutral measures. After Bonferroni correction, significant differences between left- and right-limb TFA measures were observed in 6-year-old males (t(124) = 2.96, *p* < 0.001, d = 0.53 at *p* < 0.01) and in 9-year-old females (t(110) = 2.74, *p* = 0.01, d = 0.52 at *p* < 0.01), after Bonferroni correction (Figure 1). The right limb measures were larger than those of the left limb. ICD or IMD measurements do not distinguish between limbs, as the single distance value represents measures of either genu varum or valgum, respectively. Significant differences between male and females were only observed in two- (t(35) = −2.42; *p* = 0.02, d = 0.83 at *p* < 0.05) and eight-year-olds (t(125) = −2.26; *p* = 0.03, d = −0.42 at *p* < 0.05), thus the reported means in Figure 2 were pooled per age. Figure 2 shows a peak in valgus angulation of the IMD in 3-year-olds (−2.7 cm, ±1.8 cm). Female IMDs were typically greater than those of males, except in 3-year-olds and 5-year-olds (Figure 2).

### 3.2. Quadricep-Angles (Q-Angles)

Q-angles from male and female participants were similar, although consistently larger angles were observed in females (Table 3). 

In males, statistically significant differences between left and right measures were observed in 3-year-olds (t(54) = 2.8, *p* = 0.006, d = 0.77 at *p* < 0.05); 4-year-olds (t(48) = 3.2, *p* = 0.002, d = 0.93 at *p* < 0.01); 5-year-olds (t(114) = 4.2, *p* < 0.001, d = 0.79 at *p* < 0.001); 6-year-olds (t(124) = 4.5, *p* < 0.001, d = 0.81 at *p* < 0.001); 7-year-olds (t(92) = 3.7, *p* < 0.001, d = 0.08 at *p* < 0.01) and 8-year-olds (t(98) = 3.8, *p* < 0.001, d = 0.78 at *p* < 0.001). 

In females, significant differences were observed between left and right measures in 5-year-olds (t(106) = 3.4, *p* = 0.001, d = 0.66 at *p* < 0.01); 6-year-olds (t(98) = 2.9, *p* = 0.005, d = 0.57 at *p* < 0.05); 8-year-olds (t(152) = 4.1, *p* < 0.001, d = 0.66 at *p* < 0.001) and 9-year-olds (t(110) = 3.0, *p* = 0.003, d = 0.58 at *p* < 0.05) (Figure 3). 

### 3.3. Hip Anteversion Angles (AVAs)

Combined bilateral means for AVAs of both males and females were reported (Figure 4), since no differences were noted between left- and right-limb measurements after Bonferroni correction. The greatest AVA was noted in 3-year-olds (77.6° ± 13.75°), which steadily declined with an increase in age.

### 3.4. Tibial Torsion Angles (TTAs)

Two- and 3-year-old children presented with internal tibial torsion angles (TTAs) (in-toeing), and from the age of 4 years all the children presented with neutral TTAs (Figure 5). No statistically significant differences in TTAs between males and females or between limbs (left versus right) were found, hence the means were combined. 

## 4. Discussion

This study aimed to provide profiles on LL development for black, Setswana-speaking children, between 2 and 9 years of age. Profile data of LL development in this population are required to address another study, where the LL development of back-carried children will be compared with the LL development in non-back-carried children. Thus, the novelty of this study resides within the study population, rather than the methods employed. Previous research reported differences in the development of the TFA from different ethnicities [1,4,9,10,12,15,17], but data for South African Setswana-speaking male and female children were lacking. 

We observed a similar general progression (Figure 6) in the knee angle as found by others [1,4,9,10,12,15,16,25,26]. However, the peak valgus angulations differ in peak values and the ages. We found smaller differences in the TFA between the different age groups than reported in other studies [1,3,4,8,9,10,12,15,16,17]. Clinical goniometry was mostly utilized in the studies illustrated in Figure 6; however, the Korean study [12] and European study [15] utilized radiographs and the American Caucasian [4] study used photogrammetry. The observed differences in terms of TFA development could be ascribed to either true racial differences, method, or observer-related differences between the studies. In three- to eight-year-old Greek children, researchers concluded that a physiologic valgus below 8° is present and valgus angles greater than 8° are considered as abnormal in this population [17]. In Nepalese children, a maximum valgus of 10.6° was seen in five-year-olds. Authors concluded that 11° valgus angles in one- to eight-year-old children should be regarded as abnormal [16]. We found a maximum valgus (5.6° ± 2.3°) in three-year-old Setswana-speaking children, hence any valgus TFA greater than 8° is regarded as abnormal in Setswana-speaking children. Two separate studies in Indian children [3,13] reported significant differences between males and females. Saini et al. [3] agreed with Salenius and Vanka [15] that differences in TFA development observed between male and female children are physiological [3]. We report significant differences between limbs in six-year-old males and nine-year-old females. IMD measures are also used to determine the presence of genu valgum or genu varum; however, unlike the TFA measurement, IMD cannot identify bilateral variability. Furthermore, Mathew and Madhuri [11] argued that IMD is inaccurate in comparison to TFA measurements, based on a slight alteration in standing position of the subject. Thus, TFA measurements are recommended and IMD measures should be used in addition to the TFA, and not as a standalone measure [11]. Similar IMDs to our study were reported in Nigerian children [10]. More recently, a study on TFA and IMD development in one- to eight-year-old Nepalese children found increases in IMD, and thus an increase in the valgus angulation with an increase in age [16]. Their TFA showed maximal valgus in four- to five-year-old males and five- to six-year-old females, while the IMD was at maximal in both genders in seven- to eight-year-olds. They found a strong correlation between the TFA and IMD and a fair correlation between TFA and body mass. They attributed the largest IMD observations in overweight children to soft tissue obstructions. Although not mentioned, it is possible that the older children were predominantly overweight, thus contributing to the greater IMDs observed in this age group in comparison to the TFA [16]. 

The measurements of the Q-angle and TFA are rather similar, thus arguably similar factors contribute toward the bilateral variability of both the Q-angle and TFA observed in our study. We observed an increase in the Q-angle with age and significant differences between left- and right-limb Q-angles (right larger than the left) in both males and females (Figure 3). The Q-angle bilateral variability observation is supported by other studies, in older populations [27,28]. The right limb Q-angle in athletes [27] was, unexpectedly, significantly greater than the left Q-angle. The observed differences were attributed to methodical bias, including parallax reading errors, or handedness of the investigator [27]. Similarly, in another study, the bilateral variability in Q-angle observations was attributed to a significant positive correlation found between the Q-angle and the relative lateral placement of the tibial tuberosity landmark [28]. A study investigating Q-angle development in Indian children between seven and 12 years of age reported significant positive correlations between Q-angles and age, where the Q-angle increased with age [29]. The Q-angle was measured with the participant in supine to allow comparison with an earlier study, in effect sacrificing the more functional measurement when performed in standing. In the seven- to eight-year-old Indian children, the Q-angles of female participants were larger than those of males, while in the nine- to ten-year-olds, the Q-angles were similar [29]. In a larger study, performed in active and inactive Turkish children, between nine and 19 years of age, authors aimed to determine the change of the Q-angle with age and level of activity [30]. The Q-angles in all of their youngest, nine-year-old participants were far greater than those of their oldest (19-year-old) participants [30], and also far greater than the Q-angles observed nine-year-olds in our study. Furthermore, the Q-angles of the active children were significantly lower than those of inactive children [30]. Increased quadricep strength is linked to smaller Q-angles, explaining the differences of Q-angle measures between the active and inactive groups in the Turkish study [30]. The Q-angles reported for the Indian [29] and Turkish children [30] are both greater than those in our study. The discrepancies are possibly due to our study performing measurements in standing, while other studies measured participants in the supine position [29,30]. In general, smaller Q-angles are reported for supine measurements compared to standing Q-angle measurements. Ethnic difference might therefore exist between Indian, Turkish and Setswana-speaking children. Another explanation for the negative association between age and the Q-angle found in the Turkish study [30] may be population-age discrepancies between the studies. It is possible that the Q-angle increases during the early years of life, as noted in Setswana-speaking- and Indian children, then starts decreasing after puberty. Bilateral variability was not investigated in the Indian child study, nor were significant differences found between males and females [29]. In the Turkish study, significant bilateral variability was noted in the active group, but not in the inactive group [30]. The participants in the active group were predominantly soccer players and the bilateral variability may be attributed to stronger musculature of the dominant leg versus the non-dominant leg [30]. Lastly, the normal Q-angle range in adults has been established at 15° to 20°. Similar to the findings in Turkish children, in Setswana children, the mean Q-angle was well below 15°, yet none of the children reported any musculoskeletal injuries. Thus, indicating the use of adult reference values in a child population is not appropriate [30].

Femoral AV indicates the forward torsion of the femoral neck (in the frontal plane), permitting the internal rotation of the femur and is the angle between the femoral neck axis and the transverse (mediolateral) axis of the knee [31]. An increased femoral AVA is associated with a greater range of motion for femoral internal rotation and decreased external rotation of the femur [31,32,33]. In general, the AVA in children is far greater than that in adults. In adult females between 19 and 52 years of age, normal AVAs of 12° ± 8° were observed, in comparison to 30° ± 7° observed in the same age range but in patients with increased femoral AVA symptoms [34]. Infants are born with AVAs of approximately 40°, which decreases to adult values by the age of eight to nine years old [31,32,35]. AVA stabilized and did not decrease further after eight years of age in European children [36]. In an older longitudinal study, performed on children (between four and 21 years of age) with in-toeing gait, AVA was assessed over three consecutive evaluations [33]. Overall, a reduction in the AVA with an increase in age was observed, regardless of initial increased AVA or normal alignment [33]. An increase in age indicated an association between increases in external hip rotation range of motion and a decrease in internal hip range of motion [33]. Correlations between internal- and external femoral rotation and the AVA and retroversion angle were reported in a study on 1140 children between eight and nine years old [37]. Based on the data, three groups were formed and analyzed. The majority (90%) of the sample presented with similar internal and external femoral hip rotation which did not differ more than 5° from another. A second group presented with at least 10° greater internal femoral rotation than external hip rotation. External femoral rotation was at least 10° greater than internal hip rotation in the third and smallest group (3% of the sample) [37]. A smaller sample (*n* = 147) was selected for the radiographic femoral AVA measurement. The AVA was significantly greater in group two than any of the other two groups; in comparison, the AVA was significantly smaller in the third group than the other two groups [37]. A Belgian study investigated abnormal torsional development in children [36]. The AVAs of children with increased femoral AV were similar those observed in the Norwegian study [33,36]. The increased femoral AVA group was further subdivided into an internal rotation group with internally rotated knees and feet, and an external rotation group with internally rotated knees and either neutral or external pointing feet [36]. The AVA was slightly elevated in the external rotation group in comparison to the internal rotation group, supporting the summary of Riegger-Krugh and Keysor [18]. The AVA of Belgian children was substantially lower in those with greater internal TTA than in the group with increased femoral AVAs [36]. More recently, a study on lower extremity characteristics of nine- to 18-year-old American children [38] reported lower AVAs than those found in Norwegian children [33] and far lower than those reported in European children [35]. Children were categorized into one of three groups, according to their maturation level (from the Tanner five stages of maturity index) [38]. The AVAs of females were typically greater than those of males and unlike the other studies [33,35,36], a non-linear trend was noted in both American males and females [38]. AVAs increased from the first maturation group to the second, then decreased from the second to the third group [38]. A comparison of AVA development in children between these studies [33,35,36,38] is difficult, based on participant categorization, with some values reported per age and others forming age categories or maturation categories. 

Child participants were divided into groups according to torsional problems including increased AVA and internal TTA. Increases were noted in TTAs (toward external TT) in those with increased AVA along with increased femoral medial rotation [36,37]. Decreases in TTAs (toward internal TT) were noted in those with increased internal TT, with age [36]. Significantly lower AVAs, which decreased with an increase in age, were reported in European children [31,32,35,36,37,39] in comparison to Setswana-speaking children. This might be attributed to the low validity of Craig’s test to clinically measure the AVA, as suggested by Souza and Powers [40], or an actual occurrence in the Setswana-speaking population. Craig’s test was previously found to be reliable, yet not as valid as magnetic resonance imaging (MRI) measures [40]. Greater external TTA was reported in separate studies on European- [35] and Chinese children [1], in comparison to our findings (Figure 5). Our TTAs correspond to those observed in children with increased AVAs [36], indicating a relationship between high AVA and internal TT. Without other Southern African norms, it is difficult to assess whether the greater AVA and smaller TTA measures found in Setswana children is an ethnical occurrence or owed to the clinical methods employed. 

The findings had to be interpreted against the limitations experienced with this study. Firstly, the sampling was conducted in a relatively small demographic area. Sampling from a larger demographic area would render a more accurate representation of developmental trends. However, the north-west is widespread and further north consists of rural tribal villages, which are often difficult to access. Secondly, clinical measurements are always under scrutiny for reliability and validity as well as for possible observer-related differences between studies. However, radiographic or MRI investigations are costly, especially given the large cohort in this study. Exposing the child to radiation, perhaps unnecessarily, is an ethical dilemma. 

## 5. Conclusions

In conclusion, the general development pathway of the TFA in black Setswana-speaking children was similar to those reported in European, Asian and West African children. The peak angles differed slightly and occurred at different ages than other ethnic groups, which is attributed to ethnic differences in development. ICD or IMD measures should be used in conjunction with the TFA, rather than in isolation, due to measurement inaccuracies [11]. In general, larger Q-angles were reported in children of other ethnicities in comparison to our population [29,30]. This may be due to observer-related measurement differences or errors or ascribed to either true racial differences or due to cultural practices such as back-carrying impacting the smaller Q-angles observed in Setswana-speaking children, which necessitates further research. Clinical measurement of torsion angles, especially the AVA, is often avoided, due to difficulties experienced in the reliable evaluation thereof [39,40]. We found greater AVA and smaller TTA in the Setswana population than reported for European children. We suggest a follow-up study on this population, including MRI testing of all the parameters, but especially AVA, to confirm our data. The data from this study can be used as norms for Setswana-speaking children by clinical practitioners.

## Figures and Tables

**Figure 1 ijerph-17-03245-f001:**
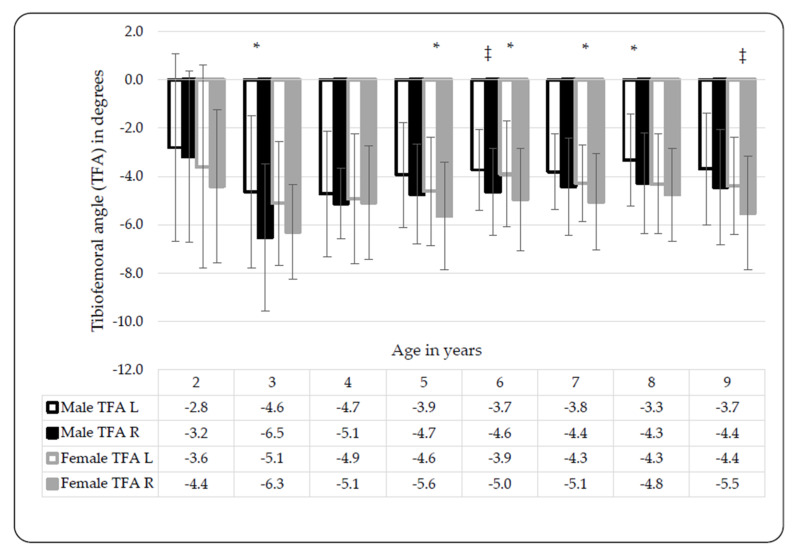
Differences in tibiofemoral angle (TFA) between black Setswana-speaking males and females; left- versus right leg. Significant differences between left and right: males: 3 years, * *p* = 0.03, d = 0.61; 6 years, ‡ *p* = 0.004, d = 0.53; 8 years, * *p* = 0.02, d = 0.49; females: 5 years, * *p* = 0.02, d = 0.46; 6 years, * *p* = 0.02, d = 0.49; 7 years, * *p* = 0.03, d = 0.43; 9 years, ‡ *p* = 0.007, d = 0.52; legend: L = left, R = right, TFA = tibiofemoral angle.

**Figure 2 ijerph-17-03245-f002:**
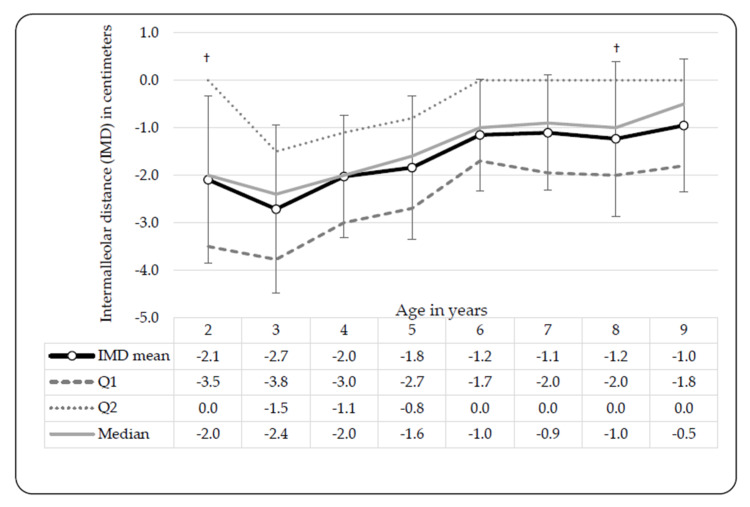
Intermalleolar distance (IMD) development in black Setswana-speaking children from age 2 to 9 years. Significant differences between males and females: 2-year-olds, † *p* = 0.02, d = −0.83; 8-year-olds, † *p* = 0.03, d = −0.42; legend: IMD = Intermalleolar distance, Q1 = 25th percentile, Q3 = 75th percentile.

**Figure 3 ijerph-17-03245-f003:**
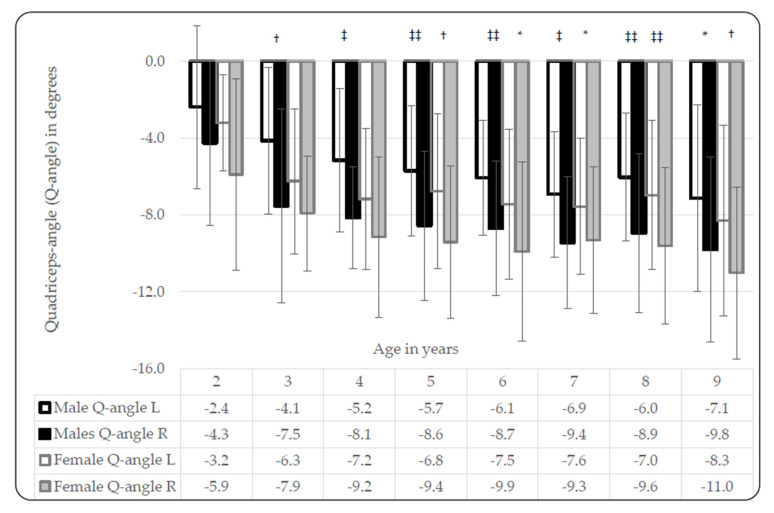
Differences in Q-angle development between black Setswana-speaking males and females; left- versus right leg. Significant differences between left and right: males: 3 years, † *p* = 0.006, d = 0.77; 4 years, ‡ *p* = 0.002, d = 0.93; 5 years, ‡‡ *p* < 0.001, d = 0.79; 6 years, ‡‡ *p* < 0.001, d = 0.81; 7 years, ‡ *p* < 0.001, d = 0.75; 8 years, ‡‡ *p* <0.001, d = 0.78; 9 years, * *p* = 0.015, d = 0.55; females: 5 years, † *p* = 0.001, d = 0.66; 6 years, * *p* = 0.005, d = 0.57; 7 years, * *p* = 0.018, d = 0.48, 8 years, ‡‡ *p* <0.001, d = 0.66; 9 years, † *p* = 0.003, d = 0.58; legend: L = left, R = right, Q-angle = quadriceps-angle.

**Figure 4 ijerph-17-03245-f004:**
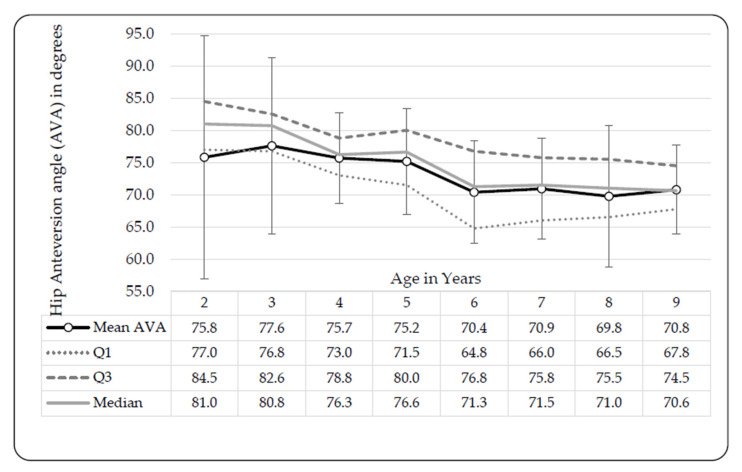
Bilateral hip anteversion angle (AVA) development in black Setswana-speaking children from age 2 to 9 years. Legend: AVA = Anteversion angle, Q1 = 25th percentile, Q3 = 75th percentile.

**Figure 5 ijerph-17-03245-f005:**
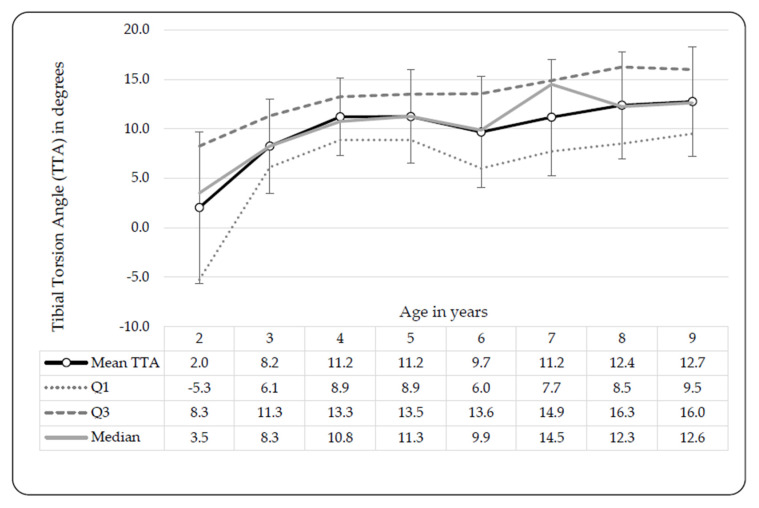
Bilateral tibial torsion angle (TTA) development in black Setswana-speaking children from age 2 to 9 years. Legend: Q1 = 25th percentile, Q3 = 75th percentile, TTA = tibial torsion angle (negative values = internal tibial torsion angle, positive values = external tibial torsion angle).

**Figure 6 ijerph-17-03245-f006:**
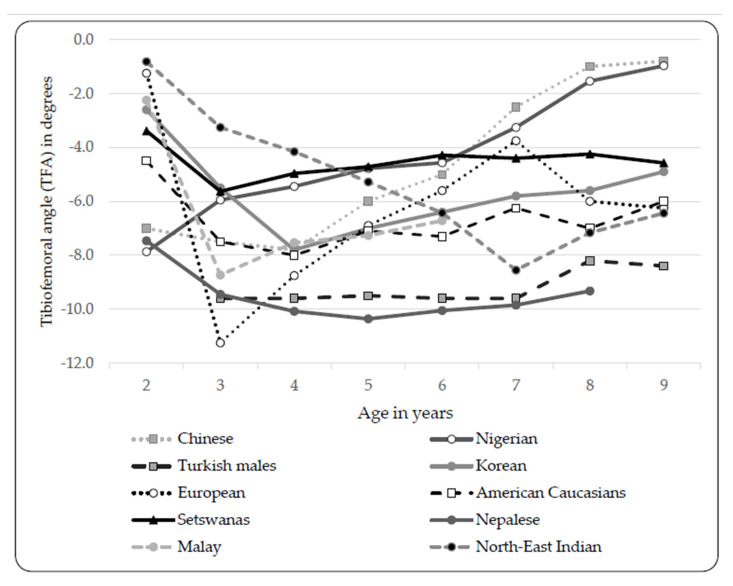
TFA development reported in children from different ethnicities [1,4,9,10,12,15,16,25,26]. Legend: TFA = tibiofemoral angle (Negative values = genu valgum, Positive values = genu varum).

**Table 1 ijerph-17-03245-t001:** Demographic characteristics of the participants per age group, in black Setswana-speaking children.

		Total					Males					Females			
Age Group	Participants *n*	Mean Age (yr) ± SD	Mean Height (cm) ± SD	Mean Body Mass (kg) ± SD	Mean BMI (kg·m^−2^) ± SD	Participants *n*	Mean Age (yr) ±SD	Mean Height (cm) ±SD	Mean Body Mass (kg) ±SD	Mean BMI ±SD	Participants *n*	Mean Age (yr) ±SD	Mean Height (cm) ±SD	Mean Body Mass (kg) ±SD	Mean BMI (kg·m^−2^) ±SD
2	38	2.10 ± 0.28	85.19 ± 7.51	12.41 ± 1.99	17.13 ± 1.92	23	2.06 ± 0.27	84.62 ± 6.13	12.23 ± 1.51	17.11 ± 1.65	15	2.17 ± 0.30	8.14 ± 9.54	12.70 ± 2.63	17.15 ± 2.36
3	52	3.00 ± 0.26	91.36 ± 6.19	14.08 ± 2.34	16.77 ± 1.95	28	2.96 ± 0.26	91.20 ± 4.94	14.41 ± 1.78	16.93 ± 2.54	24	3.05 ± 0.26	91.57 ± 7.61	13.65 ± 2.89	16.21 ± 2.17
4	54	4.06 ± 0.23	101.81 ± 5.60	16.19 ± 2.61	15.53 ± 1.34	25	4.13 ± 0.21	103.15 ± 6.44	17.09 ± 3.12	15.94 ± 1.43	29	4.00 ± 0.24	100.65 ± 4.57	15.40 ± 1.79	15.19 ± 1.18
5	112	5.08 ± 0.27	107.46 ± 5.46	18.14 ± 3.32	15.61 ± 1.75	58	5.10 ± 0.27	108.67 ± 5.74	18.75 ± 3.12	15.79 ± 1.56	54	5.05 ± 0.27	106.17 ± 4.87	17.50 ± 3.43	15.42 ± 1.92
6	113	6.06 ± 0.28	113.16 ± 5.21	20.05 ± 4.08	15.57 ± 2.34	63	6.06 ± 0.28	113.23 ± 4.79	19.80 ± 2.86	15.39 ± 1.55	50	6.06 ± 0.28	113.06 ± 5.76	20.38 ± 5.25	15.79 ± 3.08
7	98	7.01 ± 0.27	118.52 ± 6.13	21.93 ± 4.56	15.49 ± 2.01	47	6.97 ± 0.28	118.98 ± 6.44	21.85 ± 4.39	15.34 ± 2.01	51	7.04 ± 0.26	118.11 ± 5.87	22.01 ± 4.75	15.63 ± 2.03
8	127	8.07 ± 0.27	125.12 ± 6.11	26.32 ± 7.37	16.66 ± 3.59	50	8.06 ± 0.27	125.25 ± 5.78	25.61 ± 5.23	16.25 ± 2.68	77	8.07 ± 0.27	125.03 ± 6.34	26.79 ± 8.47	16.93 ± 4.06
9	97	8.92 ± 0.24	129.77 ± 6.59	27.33 ± 5.93	16.11 ± 2.61	41	8.91 ± 0.23	129.93 ± 6.76	27.45 ± 5.32	16.17 ± 2.30	56	8.93 ± 0.25	129.65 ± 6.52	27.24 ± 6.39	16.07 ± 2.84
Total	691	6.20 ± 2.03	113.56 ± 14.07	21.05 ± 6.73	16.11 ± 2.49	335	5.99 ± 2.03	112.57 ± 14.24	20.55 ± 5.81	15.96 ± 2.06	356	6.40 ± 2.00	114.55 ± 13.85	21.53 ± 7.48	16.03 ± 2.87

Legend: BMI = Body Mass Index, yr = years, cm = centimeter, kg = kilogram, kg·m^−2^ = kilogram per square meter, *n* = number of participants, SD = standard deviation.

**Table 2 ijerph-17-03245-t002:** Bilateral tibiofemoral angle (TFA) differences between males and females, per age in black Setswana-speaking children.

Age	Total					Females					Males					Significance		
	*n*	Mean	Q1	Q2	Q3	*n*	Mean	Q1	Q2	Q3	*n*	Mean	Q1	Q2	Q3	*p*-Value	Cohen’s D	Bonferroni-Corrected *p*
		±SD		(Median)			±SD		(Median)			±SD		(Median)				
2	38	−3.39	−5.44	−4.00	−1.81	15	−4.00	−6.75	−4.25	−3.13	23	−2.99	−5.13	−3.75	−1.38	0.38	−0.29	1.00
		±3.43					±3.46					±3.43						
3	52	−5.63	−7.00	−5.50	−4.25	24	−5.70	−7.06	−5.38	−4.44	28	−4.86	−7.00	−5.63	−3.81	0.84	−0.06	1.00
		±2.26					±1.83					±1.72						
4	54	−4.97	−6.25	−5.13	−3.75	29	−5.01	−6.50	−5.00	−3.75	25	−4.86	−6.25	−5.25	−4.25	0.87	−0.05	1.00
		±1.90					±2.08					±1.72						
5	112	−4.71	−5.75	−4.50	−3.50	54	−5.13	−6.00	−5.13	−4.00	58	−4.36	−5.50	−4.25	−2.75	0.04	−0.40	0.22
		±2.02					±2.01					±2.00						
6	113	−4.29	−5.25	−4.00	−3.25	50	−4.43	−5.94	−4.00	−3.25	63	−4.20	−5.00	−3.75	−3.25	0.44	−0.15	2.62
		±1.61					±1.78					±1.46						
7	98	−4.40	−5.50	−4.44	−3.25	51	−4.67	−5.75	−4.75	−3.63	47	−4.01	−5.19	−4.00	−3.00	0.08	−0.36	0.49
		±1.60					±1.56					±1.61						
8	127	−4.25	−5.38	−4.25	−3.00	77	−4.53	−5.75	−4.50	−3.25	50	−3.80	−5.00	−3.38	−2.56	0.02	−0.41	0.14
		±1.77					±1.71					±1.73						
9	97	−4.58	−6.25	−4.50	−3.25	56	−4.95	−6.75	−4.63	−3.50	41	−4.14	−5.25	−4.00	−2.25	0.04	−0.43	0.23
		±2.09					±1.99					±2.17						
Total	691	−4.51	−5.75	−4.50	−3.25	356	−4.79	−6.00	−4.75	−3.50	335	−4.22	−5.50	−4.25	−3.00	0.00	−0.28	0.00
		±2.03					±1.95					±2.07						

Legend: Q1 = 25th percentile, Q2 = 50th percentile, Q3 = 75th percentile, SD = standard deviation, TFA = Tibiofemoral angle in degrees (°).

**Table 3 ijerph-17-03245-t003:** Bilateral Q-angle differences between males and females, per age in black Setswana-speaking children.

Age	Total					Females					Males					Significance		
	*n*	Mean	Q1	Q2	Q3	*n*	Mean	Q1	Q2	Q3	*n*	Mean	Q1	Q2	Q3	*p*-Value	Cohen’s D	Bonferroni-Corrected *p*
		±SD		(Median)			±SD		(Median)			±SD		(Median)				
2	38	−3.81	−6.69	−4.88	−1.81	15	−4.55	−7.38	−5.50	−2.25	23	−3.33	−6.38	−4.00	−1.38	0.33	−0.33	1.00
		±3.77					±3.32					±4.03						
3	52	−6.41	−7.50	−6.25	−4.75	24	−7.08	−8.13	−6.50	−5.19	28	−5.84	−7.06	−5.75	−4.19	0.15	−0.40	0.90
		±3.09					±2.71					±3.31						
4	54	−7.46	−9.38	−7.25	−4.81	29	−8.16	−10.00	−8.50	−6.50	25	−6.65	−8.25	−6.25	−4.75	0.06	−0.54	0.34
		±2.92					±3.11					±2.49						
5	112	−7.59	−9.31	−7.25	−5.00	54	−8.09	−10.19	−8.13	−5.31	58	−7.13	−9.00	−6.75	−4.56	0.13	−0.29	0.81
		±3.40					±3.65					±3.10						
6	113	−7.96	−10.50	−7.50	−5.50	50	−8.68	−11.38	−8.25	−6.06	63	−7.39	−9.75	−6.88	−5.13	0.04*	−0.40	0.22
		±3.29					±3.56					±2.96						
7	98	−8.32	−10.25	−8.25	−6.31	51	−8.44	−10.38	−8.50	−6.50	47	−8.18	−10.25	−8.00	−6.13	0.66	−0.09	3.96
		±2.91					±2.94					±2.91						
8	127	−7.98	−10.25	−7.25	−5.75	77	−8.29	−10.50	−8.25	−6.00	50	−7.49	−9.50	−6.88	−5.56	0.18	−0.24	1.00
		±3.30					±3.31					±3.25						
9	97	−9.16	−12.25	−9.50	−5.75	56	−9.66	−12.31	−9.88	−6.00	41	−8.47	−11.00	−7.75	−5.25	0.19	−0.27	1.11
		±4.36					±4.27					±4.45						
Total	691	−7.74	−10.00	−7.50	−5.25	356	−8.30	−10.50	−8.25	−6.00	335	−7.14	−9.25	−6.75	−4.88	0.00	−0.33	0.00
		±3.59					±3.58					±3.51						

Q1 = 1st Quartile, Q2 = 2nd Quartile, Q3 = 3rd Quartile.

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
