# Peer review of "The Profile and Development of the Lower Limb in Setswana-Speaking Children between the Ages of 2 and 9 Years"

_ijerph, 2020, doi:10.3390/ijerph17093245_

Round 1
Reviewer 1 Report
I think that the topic is of interest and worth to be published; I have one or two maijor questions:
What is the real impact of this study; this is not very clear! There should be some aspects presented (concerning the real Impact) in the Abstract, Introduction and Discussion Parts.
What is the explanation for the differences in the different studies? This has to be discussed in a more profound way.
Author Response
Dear Reviewer, please see attached file.

Reviewer 2 Report
# Well written contents and interesting information. Some concerns should be corrected.
Introduction
# The first and second paragraphs can be united because I do not find differences between ICD, TFA, and TT or femoral retroversion. Besides, the authors should focus on the importance of each item not just describing each item.
# Abbreviations should be explained at its first presentation. LL, AVA (probably AV angle?), TTA were not described in the introduction.
# Line 61~63. the purpose should be in more detail. They should describe that they want to perform age-related and sex-related analysis for certain parameters and only age-related analysis for others.
Materials and methods
# Line 99, Anteversion angle or AVA might be better. Discard RV as the abbreviation was used only once. And the authors should suggest the version angle as quantitative value, not as a distinctive term like anteversion or retroversion.
# Table 1. Change the format to see each item should be seen by the side. And suggest each p-value. If the authors want to present bold font to significant values, all the values significant should be presented as bold.
# Statistical analysis should be described in detail. I cannot easily follow the result due to the lack of information.
Results
First, I cannot read the results easily. The authors suggest meaning, standard deviation, median, interquartile range, plus a comparison between sex for TFA, Q angle. But they did not a comparison between sex for IMD, AVA, and TTA. There is no description of why they performed statistical analysis like this.
I am confused about the tables. First of all, the age of total male and female differ significantly. Thus, the comparison between parameters could be different between total male and female groups. The total comparison may not have little importance because the male and female participants were not at the same age.
Also, the use of superscript symbols seems not appropriate. Asterix was used to suggest multiple meanings. I asked the authors to suggest the real value and make simply what is significant and not.
# Table 2. Q4 was not presented in the table. They did a t-test without normality tests even though some of the groups might not be normally distributed. (female age 2) And I cannot understand why they did Bonferroni correction comparing female TFA with male TFA. There seem to be just 2 group at each age. It seems that the authors set the null hypothesis as all age groups have the same TFA, but parameters of young ages might be changed and could not be equal.
# Figure1, What is the values in the table?
Discussion.
The discussion should be shortened. Too long to read, and the values in previous study might not be shown.
Line 210~211, I cannot agree with the notion. There can be an inter-observer difference between studies. Thus, the absolute values cannot be compared each other, and cannot conclude that racial differences exist clearly. For example, I think 2 or 3 degrees can be an observer related bias and should be considered as limitation.
Conclusion.
Discuss all parameters, not specific parameters.
Author Response
Dear Reviewer, please see the attached file. Regards, Authors

Round 2
Reviewer 2 Report
All were corrected appropriately.
1 Add the full name of TTA in line 68.
2 In each table, present the meaning of bold fonts.
Author Response
Dear Reviewer,
Thank you for raising these technical issues, we addressed them and believe the changes as suggested by you will substantially improve our manuscript.
TTA was denoted as a tibial torsion angle in line 68 (highlighted).
We opted to remove the bold font for the totals because in hindsight the totals do not add additional information and it is expected that differences would be observed in these total values.
Kind regards